# Similarities and Differences between the Orai1 Variants: Orai1α and Orai1β

**DOI:** 10.3390/ijms232314568

**Published:** 2022-11-23

**Authors:** Isaac Jardin, Alejandro Berna-Erro, Joel Nieto-Felipe, Alvaro Macias, Jose Sanchez-Collado, Jose J. Lopez, Gines M. Salido, Juan A. Rosado

**Affiliations:** Department of Physiology (Cellular Physiology Research Group), Institute of Molecular Pathology Biomarkers (IMPB), University of Extremadura, 10003 Caceres, Spain

**Keywords:** Orai1α, Orai1β, STIM1, CRAC, TRPC1, NFAT, AC8

## Abstract

Orai1, the first identified member of the Orai protein family, is ubiquitously expressed in the animal kingdom. Orai1 was initially characterized as the channel responsible for the store-operated calcium entry (SOCE), a major mechanism that allows cytosolic calcium concentration increments upon receptor-mediated IP_3_ generation, which results in intracellular Ca^2+^ store depletion. Furthermore, current evidence supports that abnormal Orai1 expression or function underlies several disorders. Orai1 is, together with STIM1, the key element of SOCE, conducting the Ca^2+^ release-activated Ca^2+^ (CRAC) current and, in association with TRPC1, the store-operated Ca^2+^ (SOC) current. Additionally, Orai1 is involved in non-capacitative pathways, as the arachidonate-regulated or LTC4-regulated Ca^2+^ channel (ARC/LRC), store-independent Ca^2+^ influx activated by the secretory pathway Ca^2+^-ATPase (SPCA2) and the small conductance Ca^2+^-activated K^+^ channel 3 (SK3). Furthermore, Orai1 possesses two variants, Orai1α and Orai1β, the latter lacking 63 amino acids in the N-terminus as compared to the full-length Orai1α form, which confers distinct features to each variant. Here, we review the current knowledge about the differences between Orai1α and Orai1β, the implications of the Ca^2+^ signals triggered by each variant, and their downstream modulatory effect within the cell.

## 1. Introduction

The calcium ion (Ca^2+^) is a ubiquitous messenger involved in several physiological events, including immune response, muscle contraction, neuronal transmission, and fertility, among others. Ca^2+^ is also heavily implicated at the cellular level, where it mediates gene expression, secretion, cell proliferation, and apoptosis. Due to its pleiotropic effects, altered intracellular Ca^2+^ homeostasis induces pathological conditions [1,2,3].

The human body possesses an intricated machinery that precisely regulates Ca^2+^ homeostasis, including the store-operated Ca^2+^ entry (SOCE), a major mechanism of Ca^2+^ mobilization in the electrically non-excitable and excitable cells, where the release of Ca^2+^ stored in the intracellular compartments, mainly the endoplasmic reticulum (ER), leads to the activation of Ca^2+^ channels in the plasma membrane (PM), followed by a massive Ca^2+^ influx from the extracellular medium [4,5]. SOCE is mediated by two types of channels, the Ca^2+^ release-activate calcium (CRAC) and the store-operated calcium (SOC) channels. The passing of Ca^2+^ from the extracellular milieu through the channels generates two distinct currents, *I_crac_* and *I_soc_*, each with identifiable features. Whereas CRAC influx is mediated exclusively by the ER Ca^2+^ sensor, STIM1, and Orai1 channels, TRPC1 proteins associate with Orai1 and STIM1 to form the SOC channels [5].

Orai1, a pivotal actor of SOCE, was first identified by three independent interference RNA screens in *Drosophila S2* cells that related a few *Drosophila* genes to SOCE, including *olf186-F*, named *Drosophila Orai* (*dOrai1*) [6,7,8]. Feske and coworkers linked their findings in *Drosophila* to a region of human chromosome 12, containing the human homolog of *dOrai1*, *Orai1* (also termed *CRACM1*), by the genetic mapping of members of a family that presented severe combined immunodeficiency (SCID). SCID patients express a homozygous R91W point mutation in the Orai1 protein that impaired T-cell activation. Over-expression of wild-type Orai1 in T cells from SCID patients reestablished SOCE [6]. In addition to SCID, gain or loss of Orai1 function has been associated with other diseases, such as autoimmune disorders (reviewed in [9,10,11,12]). Two other members of the Orai family have been identified, Orai2 and Orai3 [7], which can associate with STIM1 and STIM2 to trigger SOCE [13,14,15]. The current evidence supports that native CRAC channels consist of the heteromeric association of Orai1, Orai2, and/or Orai3 (PMID: 28294127, PMID: 29604961, PMID: 31015290), where the presence of Orai2 and Orai3 in native CRAC channels ensures that the magnitude of Ca^2+^ influx is proportional to the strength of agonist stimulation [16,17]. Despite there being no evidence indicating the composition of native CRAC channels, the stoichiometry of these channels in a given cell type presumably depends on the relative expression of Orai2 and Orai3 [15,17,18].

All three Orai proteins present the same structure: a four-membrane spanning protein containing one intracellular and two extracellular loops and the N- and C-terminus facing the cytoplasm [6,7,8] (Figure 1a). While the transmembrane domains (TM1–4) are highly conserved in all three isoforms, there are clear differences between the cytosolic regions, with 34% and 46% of sequence similarity between the N-terminus and C-terminus, respectively, of Orai1 and Orai3. The extracellular regions of the Orai proteins diverge as well, with an Orai3 third loop longer than the one exhibited by Orai1 and Orai2 (Figure 1b) [19,20,21]. In 2012, the structure of dOrai was crystallized, featuring an unexpected arrangement of six dOrai subunits conforming to the channel [22]. Although tetrameric and pentameric stoichiometries for the human Orai channels have been proposed, it is currently accepted that the Orai1 proteins might form a hexamer [21,23,24,25].

In the hexameric architecture, one dOrai channel is formed by three dimers with a crossing C-termini. The TM1 domains form the inner ring that acts as the ion-conducting pore. TM2–3 generate a central ring that protects the pore from interaction with the lipids in the PM. TM4 arranges as the outer ring, interacting with other PM components. The N-terminal sequence (aa 1–70) of human Orai1 (hOrai1) is further discussed in the following sections. An α-helix of the TM1 domain, known as the extended transmembrane Orai1 N-terminus (ETON, aa 73–90), reaches the cytoplasm and is required for the association with STIM1 and the gating of the channel [26]. The pore region is composed of a basic domain (hOrai1 aa R83, K87, and R91, whose mutation to W induced SCID), a hydrophobic segment (hOrai1 aa L95, F99, and V102), and the selectivity filter, formed by a ring of six glutamates (hOrai1 aa E106). TM1 and TM2 are connected by the first extracellular loop (loop1) that contains an acidic calcium accumulating region (CAR, hOrai1 aa D110, D112, D114) that ensures Ca^2+^ influx even in low Ca^2+^ conditions [27]. The only cytosolic Orai1 loop (loop2), located between TM2 and TM3, possesses two modulating domains that control Ca^2+^ influx through Orai1: the first one (hOrai1 aa 151–154) facilitates Orai1 fast calcium-dependent inactivation (FCDI) [28], and the second region within loop2 is an interacting domain (hOrai1 aa 157–167) with the chaperonin-containing T-complex protein 1 chaperonin complex (CCT) that mediates Orai1 internalization and recycling, thus acting as a regulator of Ca^2+^ signaling mediated by SOCE [29]. TM3 and TM4 are linked by another extracellular loop (loop3), which contains the unique glycosylation site within Orai1 (hOrai1 aa N223), allowing the interaction with lectins (carbohydrate-binding proteins) in a cell-type dependent manner, which results in the attenuation of SOCE [30]. All loops contain cysteine residues (hOrai1 loop1-C126, loop2-C143, and loop3-C195) that modulate the redox regulation of Orai1 channels as treatment with H_2_O_2_ significantly impaired SOCE [31]. The C-terminal region of dOrai extends to the cytoplasm from the TM4 domain bending in a highly conserved region (hOrai1 aa 268–291), known as TM4 extended (TM4x) [32], which results in an antiparallel association between the C-termini of two neighboring dOrai subunits [22]. This hinge region within Orai C-termini contains the initial and main binding and activating site between STIM1 and Orai1 [33,34,35,36,37]. Furthermore, Orai1 C-terminal exhibits a region (hOrai1 aa 260–275), termed as the C-terminus internalization handle (CIH), which exert a negative effect over SOCE by contributing to the internalization of Orai1 during meiosis through caveolin- and dynamin-dependent endocytic pathway [38,39].

Upon cell stimulation by physiological agonists, the reduction in the luminal ER Ca^2+^ concentration is sensed by STIM1, which suffers a conformational change that allows the activation of Orai1 channels in the plasma membrane [6,7,8,40]. STIM1 binds to Orai1 C-termini (aa 260–275) by an activating region located in the cytosolic STIM1-CC1α3 and -CC2 domain (aa 312–387) [37], which was simultaneously characterized by three independent groups and dubbed as CAD (CRAC activating domain) [34], SOAR (STIM1-Orai1 activating region) [35], and OASF (Orai1 activating small fragment) [36]. Orai1 C-termini is considered the primary binding site with STIM1, as deletion of this fragment completely abolishes the association between STIM1-Orai1 [33,41,42]. Although still a matter of debate (extensively discussed in [43]), several reports have demonstrated that STIM1 interacts with Orai1 N-termini to gate and modulate the opened Orai1 channel [26,34,44,45]. For instance, a recent study has shown that the STIM1-Orai1 N-termini interaction is vital to obtain CRAC currents featuring the hallmarks of a wild-type CRAC current [45]. A typical feature of CRAC channels is the fast inactivation mediated by Ca^2+^, which limits the number of ions passing through the channel [46]. Other proteins, such as calmodulin [47,48,49] or SARAF [50,51,52], contribute to regulating the Ca^2+^ flux through Orai1. Ultimately, when the signal responsible for intracellular Ca^2+^ stores depletion ends, STIM1 disassociates from Orai1 and returns to its coalescent state, and, subsequently, the channel closes, interrupting the Ca^2+^ influx.

Since the identification of STIM [53] and Orai [6,7,8] proteins as main players for SOCE, several models regarding the stoichiometry of both proteins have been presented [37,54,55,56,57,58,59,60,61,62,63,64]. Such knowledge is key to the creation of new drugs able to fight against conditions arising from altered SOCE [9,12]. It is clear from the beginning that ectopic expression of any of the two proteins in dissimilar ratios led to altered SOCE and CRAC currents [54,57]. In 2011, Hoover and Lewis presented the optimal STIM1:Orai1 ratio as 2:1 [58]. The identification of the dOrai1 crystal structure in 2012 [22], and further studies based on the channel architecture determined that a bimolecular coupling between 1 STIM1 dimer and 2 adjacent Orai1 subunits would suffice to activate the hexameric Orai1 channel [32,37,60,63,65,66], which did not quite fit with the most favorable STIM1:Orai1 ratio [58]. However, new pieces of evidence have been presented against this bimolecular model, favoring a unimolecular association between 1 STIM1 dimer and 1 Orai1 subunit [62,64,67], which fits with the 2:1 optimal ratio. Zhou and coworkers proposed that the formation of the pocket by to Orai1-TM4 through I316 and I319 may be an artifact of dOrai1 crystallization conditions; thus, it would be improbable that 1 STIM1 dimer might bind to and activate 2 neighboring Orai1 subunits [67]. Undoubtedly, this subject requires further investigation, and probably, we will not solve all the questions until the full hOrai1 structure is identified. Fortunately, new technologies, such as neural networks or artificial intelligence systems capable of predicting a protein’s 3D structure from its amino acid sequence [68], might shed some light, helping us to unravel the mystery.

In addition to the Ca^2+^ influx through CRAC channels, Orai1 is also responsible for the Ca^2+^ entry mediated by the SOC channels in collaboration with STIM1 and TRPC1 [5,69,70,71,72]. STIM1 activates TRPC1 via the association of a region of STIM1 C-termini polybasic domain (aa 684–685) with two conserved aspartates (aa 639–640) within TRPC1 [70,71]. The activation of SOC channels via receptor-mediated inositol 1,4,5-trisphosphate (IP_3_) and the subsequent Ca^2+^ store release generates characteristic changes in cytosolic Ca^2+^ concentration, where Orai1 conducts the generation of Ca^2+^ oscillations and TRPC1 is implicated in the frequency of baseline Ca^2+^ oscillations, supporting, as well, a maintained Ca^2+^ elevation with higher agonist concentration [72,73]. Orai1-mediated Ca^2+^ oscillations control NFAT translocation to the nucleus and NFAT-dependent gene expression, probably due to the proximity of NFAT machinery neighboring the pore of Orai1 [72,74,75]. While TRPC1 have no implications in NFAT activation [72], it is involved in the activation of NFκB and the NFκB-dependent gene expression [72,76,77]. Furthermore, TRPC1 Ca^2+^ entry upon STIM1 activation is crucial for triggering epithelial-to-mesenchymal transition (EMT) in invasive ductal carcinoma breast cancer cells, where SOCE is mainly mediated by TRPC1 channels, and the implication of Orai1 channels is still unknown [78]. Additional studies are needed to understand the physiology of SOC channels, which will help us to finally explain their implication in the pathophysiology of certain diseases, such as cancer.

Moreover, Orai1 supports other non-capacitative Ca^2+^ entry pathways, as well. For instance, Orai1 associates with STIM1 and Orai3 to form the arachidonate-regulated or LTC4-regulated Ca^2+^ channel (ARC/LRC) [79,80]. Furthermore, Orai1 might act independently of STIM1 associated with the secretory pathway Ca^2+^-ATPase-2 (SPCA2) [81,82] or the small conductance Ca^2+^-activated K^+^ channel 3 (SK3) [83]. Store-independent Ca^2+^ entry and its role in physiology and pathophysiology are extensively reviewed in [84,85].

Orai1 is a ubiquitous and complex Ca^2+^ channel that contributes to several local and global Ca^2+^ signals, which modulate the downstream effect of many Ca^2+^-dependent pathways. Alterations in those routes lead to diseases [12], thus, a thorough understanding of Orai1 is required. Recently, two Orai1 variants have been identified [5], which might explain some of the questions that we had but have raised several new ones. Here we review the current knowledge about the Orai1 variants and their implication in Ca^2+^ signals mediated by Orai1.

## 2. Orai1 Variants: Orai1α and Orai1β

Orai proteins are evolutionarily conserved across the animal kingdom. Analysis of the phylogenetic relationships of the Orai proteins in different species has revealed that invertebrates contain single copies of the Orai protein (except for *Tribolium castaneum* and *Apis mellifera,* which contain two Orai molecules). The Orai protein found in invertebrates evolved into two Orai proteins in vertebrates, referred to as Orai1 and Orai2 in mammalian cells. In mammals, specifically, duplication of the *Orai1* gene led to Orai3, which exhibits closer phylogenetic relationships with Orai1 [86].

At present, no variants of Orai1 derived from alternative pre-messenger RNA splicing have been described; however, two forms of Orai1 have been identified in mammalian cells generated by alternative translation initiation. The human full-length Orai1 variant, named Orai1α, contains 301 amino acids (Figure 2a), while the short form, known as Orai1β (Figure 2b), arises from alternative translation initiation at methionine 64, or even methionine 71, therefore lacking the N-terminal 63 or 70 amino acids present in Orai1α [87,88]. It has been reported that the Kozak sequence for the first methionine in the native 5′-untranslated region is rather weak, leading to the translation of Orai1β from an alternative translation start site, methionines 64 and 71. Accordingly, improvement of the Kozak sequence for the first methionine has been reported to lead to the generation of Orai1α exclusively, as well as mutation of methionines 64 and 71 to alanine or valine [87]. Conversely, mutation of the first methionine results in the production of Orai1β, which strongly suggest that methionines 64 and 71 function as second translation initiation sites in Orai1 [88].

Different functional domains have been identified in the Orai1α exclusive N-terminal region, containing 63 amino acids, which have been associated with the sensitivity to Ca^2+^-dependent inactivation, protein-protein interaction, or cellular location. Specifically, residues 26–34 in Orai1α are involved in its interaction with adenylyl cyclase 8 (AC8), a Ca^2+^-modulated cyclase with low affinity for this cation [89]. In addition, Orai1 has been reported to be phosphorylated by PKC at N-terminal serines 27 and 30 residues, an event that is strongly associated with the suppression of SOCE and CRAC channel function [90]. Furthermore, serine 34 is a PKA phosphorylation site reported to mediate Ca^2+^-dependent Orai1 channel inactivation as a feedback mechanism upon the activation of AC8 [74]. There is also a predicted PIP2-binding domain between amino acids 28–33 whose functional role is still uncertain but might be involved in the distribution of Orai1 in the plasma membrane [91] or the regulation of STIM1-Orai1 interaction by the protein SARAF [92]. Finally, residues 52–60 form a predicted caveolin-binding domain which might regulate the plasma membrane location of Orai1 as it has been reported during meiosis, where Orai1 internalization has been shown to be dependent on its interaction with caveolin [39].

Orai1α and Orai1β are expressed in all the cells investigated, including HEK-293 and HeLa cells, as well as a variety of tumor cells, such as luminal MCF7 and triple-negative MDA-MB-231 breast cancer cells, squamous carcinoma A431 cells, lung adenocarcinoma H441 cells and colorectal adenocarcinoma T84 cells [87,93]. The Orai1α:Orai1β expression ratio varies among the cell lines investigated but mostly ranging from 0.3 to 1 [87,93], thus suggesting that Orai1 mRNA transduction mostly favors the expression of the short variant.

Analysis of Orai1α and Orai1β subcellular locations has revealed that, as expected, both Orai1 variants are predominantly localized in the plasma membrane [87]. Figure 3 shows that Orai1α-GFP and Orai1β-GFP expressed in Orai1-KO HEK-293 cells are exclusively located at the plasma membrane; therefore, the truncation of N-terminal 63 amino acids in Orai1β does not affect its plasma membrane location in resting cells. Furthermore, Fukushima and coworkers demonstrated that Ca^2+^ store depletion using the SERCA inhibitor thapsigargin or the physiological agonist carbachol resulted in the accumulation of Orai1α and Orai1β into puncta with similar efficiency, thus indicating that Orai1α and Orai1β exhibit a similar subcellular location [87]. Nevertheless, although no differences have been found in the cellular distribution of both Orai1 variants, the mobility of Orai1α and Orai1β in the plasma membrane significantly differs. As analyzed by rates of fluorescence recovery after photobleaching (FRAP), either expressed singly or together, the half-time of Orai1β recovery follows a normal distribution while that for Orai1α exhibits a bimodal distribution, the first one with a half-time similar to that of Orai1β followed by a second, predominant, population with a slower rate of recovery. The nature of the two populations of Orai1α rates of recovery remains uncertain. The observation that Orai1β exhibits the same mobility profile either expressed alone or co-expressed with Orai1α, which is different from the Orai1α mobility profile, suggests that Orai1α and Orai1β do not form heteromeric channels; however, further studies are required to characterize the two Orai1α populations and whether the minor, faster, population of Orai1α might heteromerize with Orai1β under certain conditions, as co-expression of both Orai1 forms slows down their rates of recovery [87].

## 3. Functional Properties of Orai1α and Orai1β

Using Orai1α- or Orai1β-optimized constructs that results in the production of one of the Orai1 forms, Desai and coworkers demonstrated that both Orai1 variants can rescue SOCE in Orai1-KO mouse embryonic fibroblasts (MEFs), with Orai1α showing smaller efficiency. Similarly, expression of Orai1α or Orai1β in Orai1-KO MEFs and HEK-293 cells restored *I*_crac_ but with similar efficiency [5]. The reason for the discrepant efficiencies of both Orai1 variants in SOCE and *I*_crac_ was attributed to the dialysis of the intracellular fast Ca^2+^ chelator BAPTA for *I*_crac_ measurements, which impairs Ca^2+^-dependent inactivation of the Orai1 channels. In fact, analysis of fast Ca^2+^-dependent inactivation of Orai1α and Orai1β expressed in HEK-293 cells revealed that, in the presence of the slower Ca^2+^ chelator EGTA, Orai1α produced a smaller current than Orai1β at all potentials tested, an observation that was abrogated in the presence of BAPTA. These findings indicate that, while both variants support *I*_crac_, Orai1α shows a stronger fast Ca^2+^-dependent inactivation than Orai1β, the latter being barely sensitive to Ca^2+^-depending inactivation as it behaves similarly in the presence of EGTA and BAPTA [5]. These findings were confirmed in Orai1-KO HEK-293 cells expressing Orai1α or Orai1β using plasmids with the weak thymidine kinase (TK) promoter to reconstitute Orai1 expression at a more physiological level [74]. Zhang et al. also reported that the substitution of extracellular Ca^2+^ by Ba^2+^ abrogates the differences between Orai1α and Orai1β, which strongly suggests that the different inactivation rates are mediated by Ca^2+^ [74].

The functional role of Orai1 variants in the *I*_soc_ current has also been investigated. This current is the result of a complex mechanism where Ca^2+^ influx through Orai-forming CRAC channels recruits TRPC channels to the plasma membrane [94]; therefore, *I*_soc_ comprises the currents mediated by CRAC and TRPC1 channels [5,94]. In HEK-293 cells, patch-clamp experiments have demonstrated that both Orai1α and Orai1β support *I*_soc_ with similar efficiency [5]. These findings were confirmed by analyzing Mn^2+^ influx through TRPC1 channels [73]. Mn^2+^ is used as a surrogate for Ca^2+^ that enters the cells through TRPC1 but not via Orai1 channels in the presence of extracellular divalent cations, such as Ca^2+^ [40,73]. However, the involvement of Orai1β in this mechanism seems to be cell-specific, as, in HeLa cells, we have found that Orai1α, but not Orai1β, participates in the recruitment of TRPC1 channels in the plasma membrane and is required for store-operated cation entry through TRPC1. In this cellular model, Orai1β was not found to interact or co-localize with TRPC1 channels, and expression of a dominant negative Orai1β mutant failed to attenuate translocation of TRPC1 to the plasma membrane or Mn^2+^ influx through TRPC1 channels [73]. The basis for the different involvement of Orai1β in TRPC1 function and *I*_soc_ in these cell types is uncertain at present.

One of the most relevant functional differences between both Orai1 forms is that Orai1α, but not Orai1β, has been shown to support *I*_arc_ [5], the store-independent Ca^2+^ selective current activated by arachidonate and its metabolite leukotriene C4 [95]. ARC channels consist of heteropentamers of three Orai1 subunits and two Orai3 subunits [20,79] and require the participation of plasma membrane resident STIM1 [80,96]. In HEK-293 cells, individual expression of Orai1α was found to rescue *I*_arc_ in cells transfected with siOrai1, while Orai1β failed to do this.

## 4. Orai1 Variants and NFAT Activation

Nuclear factor of activated T-cell (NFAT) proteins are widely expressed transcription factors that regulate a plethora of genes involved in immunity, development, and, more recently, having an important function in cancer physiology [97,98]. Four out of five members NFAT(1–4) are activated by phosphatases that respond to rises in cytosolic free-Ca^2+^ concentration ([Ca^2+^]_i_), while only NFAT5 is sensitive to osmotic stress [99]. Ca^2+^-sensitive NFATs are mainly activated by Orai1/CRAC channels [98,100], but increasing evidence supported its activation by other channels involved in different modalities of Ca^2+^ influx into the cell, such as L-type [101,102,103], T-type [104,105,106] and members of the TRP family not related with SOCE [100,107,108,109,110,111,112,113]. The amplitude, duration, oscillatory pattern, and source of Ca^2+^ transients modulate the later activation of NFAT [15,16,114,115,116,117]. Thus, raises of [Ca^2+^]_i_ promoted by opened PM Ca^2+^-channels activate the Ca^2+^-dependent calmodulin (CaM), which contains EF-hand motifs to sense [Ca^2+^]_i_, among other proteins. As a result, heavily phosphorylated resting NFAT proteins located in the cytoplasm become dephosphorylated by calcineurin, starting the calcineurin-NFAT pathway [99]. Active dephosphorylated NFAT migrates towards the nucleus, binding there to the promoter region of many genes, both alone or forming complexes with other transcription factors (for instance, AP-1, GATA-4, or MEF-2) [99]. It has been suggested that calcineurin remains bound to NFAT to prevent rephosphorylation during translocation [118]. Rephosphorylation on nuclear NFAT by nuclear kinases (such as GSK3, JNK, p38) implies its inactivation and relocation to the cytoplasm [99]. This way, PM Ca^2+^-channels can trigger gene transcription through the calcineurin-NFAT pathway.

Focusing on SOCE, Orai1-mediated *I_crac_* trigger NFAT(1, 2 and 4) translocation [6,15,74,75,103,117,119,120,121], while its participation in NFAT(3 and 5) translocation has not been yet reported. However, NFAT5 has been proposed as a powerful regulator of Orai1 expression in megakaryocytes, the precursor cells of blood platelets [122]. As aside, Orai2 and Orai3 are not able (or less able) to activate NFATs translocation themselves in the absence of Orai1 [16,75,119]. Nevertheless, their presence can modulate Orai1-dependent NFATs activation by heteromerization with Orai1 [15,16,17,18,123,124], despite their inability to directly activate NFATs. Therefore, Orai2 and Orai3 seem to act as Orai1 modulators rather than NFAT1 activators [15,16].

Since the identification of Orai1α and Orai1β variants has only recently been reported, previous studies mostly described Orai1α-NFAT interactions. The role of the Orai1α subunit in calmodulin and NFAT1 activation is not only restricted to the generation of local high [Ca^2+^]_i_ microdomains within ER-PM junctions. The Orai1α region located within the amino acid residues 39–59 (AKAR region) interacts directly or indirectly with the scaffolding protein A-kinase anchoring protein 79 (AKAP79) [75]. Once complexed with Orai1α through AKAR, AKAP79 binds calcineurin and NFAT together, facilitating their interaction, the calcineurin-mediated NFAT1 dephosphorylation, and the subsequent NFAT activation [75,119]. In conclusion, Orai1α not only promotes the initiation but also compartmentalizes the complexed AKAP79/calcineurin-NFAT pathway into high [Ca^2+^]_i_ microdomains during SOCE.

The Orai1α-AKAP79 interaction seems to be specific since the AKAR region is absent in Orai2 or Orai3 subunits, and they are less able to interact with AKAP79 [75]. Interestingly, Orai1β also lacks this N-terminal region necessary to form a complex with AKAP-79, but the impact of its absence on NFAT activation is controversial. Kar and coworkers proposed that Orai1-AKAP79 interaction is mandatory to drive the activation of NFAT translocation in HEK-293 cells. They reported a relatively weak Orai1β-AKAP79 interaction due to the absence of the AKAR region that, despite the strong Ca^2+^-influx promoted by Orai1β, is too weak to promote NFAT1 translocation to the nucleus. Thus, Orai1β might fail to trigger NFAT1-dependent gene expression due to the lack of the N-terminal region. This idea is reinforced by the fact that Orai2 and Orai3 also lack AKAR, resulting in much less NFAT1 trafficking to the nucleus. However, it might indicate a remaining residual mechanism able to promote NFAT1 dephosphorylation and translocation to a lesser extent [75]. In summary, Kar et al. suggest that the presence of Orai1β might negatively contribute to Orai1α -mediated NFAT1 activation, comprising then a mechanism to suppress gene expression without interfering with other SOCE functions [75]. Given that, this study implies that raises in [Ca^2+^]_i_ themselves are not sufficient and that it is mandatory for the formation of Orai1/AKAP-79/calmodulin/NFAT1 complexes to drive NFAT1 translocation.

AKAP79 also binds and recruits NFAT4 near L-type channels in neurons [103]. By extrapolation, one might expect an abrogated Orai1β-NFAT4 interaction as well; however, Zhang and coworkers have reported direct NFAT4 activation by Orai1β also in HEK-293 cells [74]. Zhang et al. show that both Orai1α and Orai1β are equally able to activate NFAT1 [74]. Both variants activate NFAT1 and NFAT4 translocation at high agonist concentrations, but only Orai1β can trigger NFAT4 translocation at physiological agonist concentrations. The proposed mechanism explains how different [Ca^2+^]_i_ oscillatory frequencies modulate alternative NFAT1/NFAT4 patterns of translocation, which were previously observed using different agonist concentrations [114,117,125,126]. Thus, authors reported that Orai1β are less sensitive to FCDI as previously reported [5,87], which will be important at the end for NFATs modulation. FCDI attenuates *I_crac,_* and it is characteristic of Orai1α [74]. Ca^2+^-dependent inactivation (CDI) comprises two components; a fast inactivation (FCDI) occurring within milliseconds that is promoted by the high [Ca^2+^]_i_ microdomain formed near the channel, and a slow inactivation (SCDI) occurring over seconds triggered by overall rises in [Ca^2+^]_i_ [127]. CDI was previously proposed as a mechanism to avoid toxic [Ca^2+^]_C_ rises, but this work also proposes that FCDI modulates the frequency of [Ca^2+^]_C_ oscillations. The lesser sensitivity to CDI exhibited by Orai1β has been attributed to the absence of Ser34, which, in Orai1α, is phosphorylated by protein kinase A (PKA), leading to CDI initiation. The lack of CDI entails the promotion of different oscillatory [Ca^2+^]_i_ frequencies by Orai1β than those generated by Orai1α. Thus, Orai1α leads to lower frequency [Ca^2+^]_i_ oscillations than Orai1β, which might underlie the activation of NFAT isoforms at different degrees depending on the agonist concentration [74,117]. Both Orai1 isoforms generate robust *I_crac_* at high agonist stimulation, and both would activate NFAT1 and NFAT4, but only Orai1β would activate NFAT4 and not NFAT1 because NFAT1 does not respond to weak Ca^2+^-currents, at low physiological agonist stimulation. The translocation dynamics of both activated NFATs are different. Active NFAT4 translocates faster and remains transiently, in an oscillatory mode, inside the nucleus, while NFAT1 remains in a more sustained manner [15,125,126]. This mechanism seems to have a similar goal to the heteromerization of Orai2 or Orai3 with Orai1, the generation of different patterns of NFAT1/NFAT4 activation [15,16], except that the Orai1α/Orai1β heteromerization has not been demonstrated [87]. Although the work suggests that Orai1- and IP_3_R-dependent Ca^2+^-currents are important for NFATs activation, it does not clarify which one is the determinant one. Other studies reported that NFAT4 requires Ca^2+^ influx from two sources, from PM channels and nuclear IP_3_R, to be activated [117,126]. Nevertheless, their hypothesis implies that raises in [Ca^2+^]_i_ are sufficient to activate NFATs in contrast to Kar and coworkers and that their oscillatory nature modulates NFATs function. Regarding other NFAT isoforms, there are no studies about the possible interactions between Orai1β and NFAT2 or NFAT3. Orai1α activity triggers NFAT2 translocation [121]. Whether the interaction is mediated by AKAP79 or the NFAT2 is unresponsive to Orai1β activation remains unsolved.

## 5. Regulation of Orai1α by AC8 in Breast Cancer Cells

Mammalian adenylyl cyclases comprise nine (AC1–9) transmembrane enzymes and one cytosolic isoform (AC10) that catalyze the synthesis of cyclic AMP (cAMP) from ATP [128]. AC isoforms are located throughout the organism, displaying specific locations and presenting cell and tissue-specific expression patterns [129]. ACs are classically activated by G-proteins downstream G-protein coupled receptors, but additionally, AC1 and AC8 are activated by Ca^2+^ [129,130,131,132,133,134,135,136,137]. cAMP is a basic second messenger that has a major function in cellular physiology. Molecular components of cAMP signaling pathways are highly intracellularly compartmentalized and trigger a wide spectrum of basic cellular functions [138]. PKA, for instance, is one of the cAMP effectors and controls a broad spectrum of downstream signaling cascades [139]. Target molecules of PKA differ depending on where this kinase is subcellularly located. Its compartmentalization is regulated by scaffold proteins that form complexes with PKA to anchor it close to target molecules. Several studies demonstrated that SOCE or SOCE components can trigger AC5, AC6, and AC8 activation [130,131,132,133,134,135,136,137]. AC8 is regulated by Ca^2+^ and exhibits a low affinity for this ion. This isoform is able to form complexes with Orai1 channels [89,140], both proteins co-localize into lipid rafts [141], and residues 26–34 of the Orai1α N-terminal region [89] directly bind to the N-terminal region of AC8 [142]. The Orai1α region also contains the AKAR region, which is necessary to bind to AKAP79, which recruits PKA. Palmitoylation targets AKAP79 to lipid rafts where Orai1 and AC8 are located [143]. Thus, the N-terminal region of Orai1α recruits AC8/AKAP79/PKA closer, assembling the cAMP signaling complex nearby.

The Orai1-induced AC8 activation triggers a signaling loop ending in Orai1α phosphorylation by PKA. Orai1α-mediated SOCE activates AC8 in HEK293 cells overexpressing the latter, generating local microdomains of high cytoplasmic cAMP concentration ([cAMP]_c_), that activates the complexed PKA, which, in turn, phosphorylates Orai1α at Ser34. The consequence of Ser34 phosphorylation is the activation of CDI since the mutation of this phosphorylation site (the exchange of a Ser34 by Arg34) abrogates CDI [74]. cAMP signaling is later degraded by phosphodiesterases to finish the signaling pathway [144,145]. Interestingly, Ser34 is also phosphorylated by PKG, having similar results on Orai1 activity [146]. Two additional Ser residues (Ser27 and Ser30) phosphorylated by PKC were described within the N-terminal region, whose function is to reduce Orai1 channel activity [90]. Interestingly, those three phosphorylatable Ser residues are located within the 26–34 residue region that contains the AC8 binding site [89]. Thus, the AC8-dependent cAMP signaling pathway comprises a system to trigger CDI after Orai1 channel activation [74]. By contrast, Orai1β does not bind to AC8 [74,93] due to the lack of the N-terminal region containing both AKAR and AC8 binding regions [74]. Orai1β is, therefore, unable to bind and recruit AKAP-79 and AC8, failing to recruit the AC8/AKAP-79/PKA complex responsible for cAMP-dependent signaling. This interaction, therefore, seems to be selective to Orai1α since neither Orai2 nor Orai3 contain both AKAR and AC8 binding regions as well [89]. As a result, Orai1β is less sensitive to CDI, as depicted in the previous section. The presence/absence of CDI after Orai1 channel activation would shape the amplitude of SOCE and the pattern of Ca^2+^ oscillations, establishing different NFATs translocation patterns [74]. AC8 participates in CDI since the Orai1α mutant lacking the AC8 binding site still displays a diminished CDI [74]; however, AC8 is not the only inducer of CDI; for instance, SARAF was previously proposed as a regulator of CDI after Orai1 channel activation [147]. In agreement, previous studies revealed that cAMP increases attenuate SOCE [148,149,150]. Hypothetically, the absence of the cAMP signaling complex close to Orai1β-assembled channels might have consequences in Ca^2+^ signaling. It is known that cAMP modulate Ca^2+^ homeostasis (extensively discussed in [136]). In summary, the high [cAMP]_c_ microenvironment generated around activated Orai1α subunits regulates not only *I_crac_* but also other Ca^2+^-handling mechanisms. Therefore, the distinct ability of Orai1 variants to activate AC8 might differentially shape the Ca^2+^ signals in cells where this cyclase plays a functional role.

It has been reported that embryonic kidney HEK293 cells do not express AC8, but they can recruit AKAP79/PKA and phosphodiesterase 4 (PDE4) complexes nearby activated Orai1 channels in the absence of cAMP microdomains suggesting that there should be an alternative source of cAMP able to trigger PKA activity close to Orai1 [145].

In cancer cells, the cAMP/PKA pathway promotes proliferation, migration, and invasive properties, as well as other aspects of their physiology [151]. Moreover, the Orai1 channel is a well-known regulator of proliferation and migration in breast cancer cells [136,152]. Therefore, the SOCE-dependent cAMP signaling might have increased importance in breast cancer biology since both Orai1 and AC8 have been found to be highly expressed in these cells [93,153,154,155,156]. This is consistent with previous evidence supporting a remodeling of the Ca^2+^-signaling machinery in cancer [136]. Indeed, both Orai1α and Orai1β were found to be highly expressed in triple-negative breast cancer MDA-MB-231 and luminal MCF7 cell lines [93]. In these cells, AC8 overexpression interferes with PKA, PKC, or PKG-induced Orai1 inactivation, impairing Ser27, 30, or 34 phosphorylation as these residues overlap with the AC8 binding motif at the N-terminal region of Orai1α subunit. AC8 is predominantly overexpressed over Orai1, and AC8 interaction has been reported to restrict the accessibility to the phosphorylatable Ser residues. The consequence is that AC8 enhances Orai1-mediated SOCE in breast cancer cells attenuating Orai1α CDI [136]. Since Orai1β does not interact with AC8, it remains unaffected. Moreover, it has been reported that AC8 knockdown attenuates cell proliferation and migration in breast cancer cell lines, in contrast to non-tumoral breast epithelial cell lines that express lesser AC8, offering a molecular explanation on previous observations reporting a role of Orai1-mediated SOCE in proliferation and migration in breast cancer cells [136,152]. Finally, the transcription factor cAMP response element-binding protein (CREB) is a well-known substrate of PKA. It has been reported that SOCE can also activate CREB [157,158,159], and the induction of CREB activity by AC8 has been demonstrated in neurons [160,161]. Overstimulation of CREB has been associated with cancer [132]. Therefore, Orai1α-AC8 interaction might link SOCE with CREB function, allowing SOCE to modulate CREB-dependent gene expression [162].

## 6. Conclusions

Orai1 is the key pore-forming protein of the CRAC channels, which mediate the prototypical and best-characterized store-operated current *I*_crac_. In addition, Orai1 promotes store-dependent activation of the *I*_soc_ current involving TRPC1. Two variants of Orai1 are expressed at the protein level in mammalian cells. These variants, Orai1α and Orai1β, generated by alternative translation initiation, differ in the N-terminal 63 amino acids lacking in the short form, Orai1β. While these forms have been reported to support *I*_crac_ with similar efficiencies, their participation in *I*_soc_ depends on the cell type investigated, and *I*_arc_ activation is unique to Orai1α. Orai1 variants also differ in their sensitivity to fast Ca^2+^-dependent inactivation, which explains the different Ca^2+^ signals mediated by these forms when expressed individually. The absence of the N-terminal 63 amino acids in Orai1β limits the interaction with different partners, including AC8, AKAP79, or caveolin, as well as serine phosphorylation at residues 27, 31 and 34, present in Orai1α. These differences have been proposed to underlie the distinct channel inactivation properties, mobility profiles, and NFAT activation mechanisms. The presence of Orai1α and Oraiβ in a given cell type provides an additional tool to generate Ca^2+^ signals appropriate to the intensity of agonist stimulation. 

## Figures and Tables

**Figure 1 ijms-23-14568-f001:**
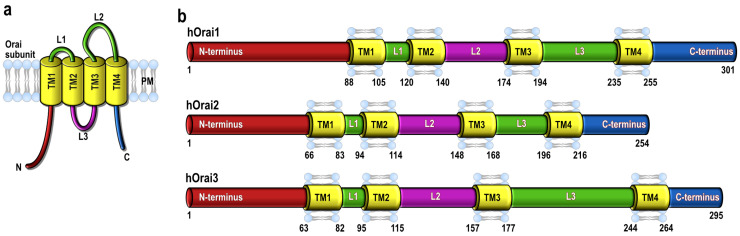
(**a**) Predicted topology of Orai proteins. (**b**) A schematic representation of the three members of the Orai family with their structural domains. Abbreviations: L, loop; TM, transmembrane domain.

**Figure 2 ijms-23-14568-f002:**
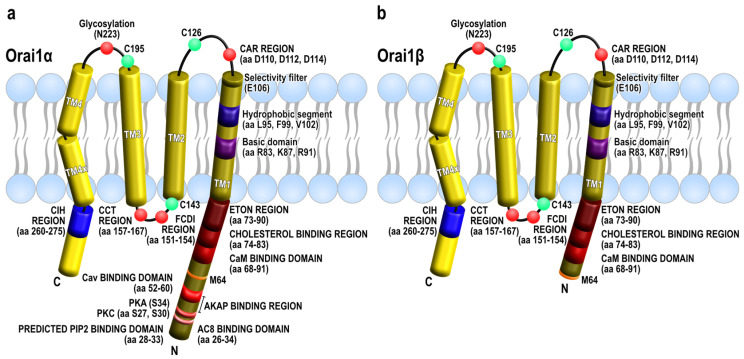
Cartoon depicting Orai1α or Orai1β relevant domains. Both variants of Orai1, Orai1α (**a**) and Orai1β (**b**), are presented with several key amino acids and domains required for their function and the association with other proteins. Abbreviations as they appear from the N– to the C–termini: AC8, adenylyl cyclase 8; PIP2, phosphatidylinositol bisphosphate; PKC, protein kinase C; PKA, protein kinase A, AKAP, A–kinase anchoring protein; Cav, caveolin; CaM, Calmodulin; ETON, extended transmembrane Orai1 N-terminal region; CAR, Ca^2+^ accumulating region; C126, C143 and C195, cysteine residues that modulate the redox regulation of Orai1; FCDI, fast Ca^2+^-dependent inactivation; CCT, chaperonin–containing T–complex protein 1 chaperonin complex; CIH, C-terminus Internalization Handle; TM, transmembrane domain; TM4x, extended transmembrane domain 4.

**Figure 3 ijms-23-14568-f003:**
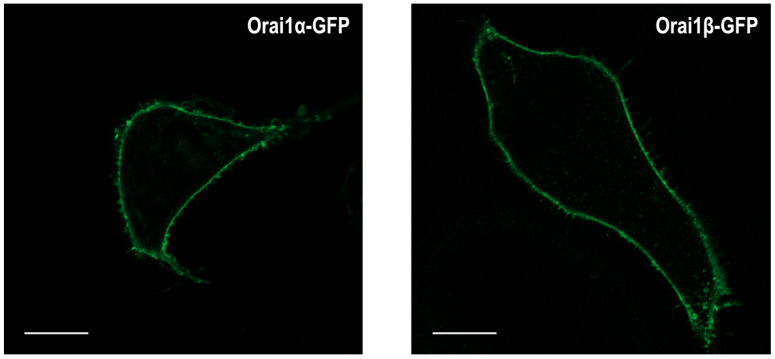
Orai1α or Orai1β subcellular location. Representative confocal images of Orai1α-GFP or Orai1β-GFP expressed in Orai1-KO HEK-293 cells. Scale bar: 10 µm.

## Data Availability

Not applicable.

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
