# Peer review of "Similarities and Differences between the Orai1 Variants: Orai1α and Orai1β"

_ijms, 2022, doi:10.3390/ijms232314568_

Round 1
Reviewer 1 Report
This manuscript is a well written and understandable review paper. The authors interpreted the relevant results correctly. The implication from this review may give readers some ideas to their own research.
I have one recommendation and one correction below:
1. The figure 1 cannot help readers to fully understand the sentence of "All three Orai proteins present the same structure: a four-membrane spanning protein containing one intracellular and two extracellular loops, and the N- and C-terminus facing the cytoplasm" at line 62. I wish the authors can provide another figure here, otherwise this sentence needs to be rewritten.
2. On the line 23, 64 amino acids should be changed to 63 amino acids.
Author Response
Dear Editor and Reviewer,
Thank you for your letter dated 14th November 2022 concerning the above manuscript. We would like to thank the Reviewers for their comments in order to improve the quality of the manuscript. We have modified the manuscript introducing all the requirements and suggestions of the Reviewers. Our responses to the points raised and the modifications made are detailed below.
Yours sincerely
Dr. Juan A. Rosado,
Reviewer #1:
This manuscript is a well written and understandable review paper. The authors interpreted the relevant results correctly. The implication from this review may give readers some ideas to their own research.
Answer: We thank the Reviewer for his/her supporting comments.
I have one recommendation and one correction below:
- The figure 1 cannot help readers to fully understand the sentence of "All three Orai proteins present the same structure: a four-membrane spanning protein containing one intracellular and two extracellular loops, and the N- and C-terminus facing the cytoplasm" at line 62. I wish the authors can provide another figure here, otherwise this sentence needs to be rewritten.
Answer: We appreciate the Reviewer´s comment. As suggested, we have introduced a new figure (Figure 1) depicting the predicted topology of Orai proteins and a schematic representation of the three members of the Orai family with their structural domains. We have accordingly renumbered the other two figures as Figure 2 and Figure 3, respectively.
- On the line 23, 64 amino acids should be changed to 63 amino acids
Answer: We thank the Reviewer for drawing our attention to this point. We have corrected this mistake.
Reviewer 2 Report
Comments
Line 58: “the relevance of ORAI2 and ORAI3 remains still unclear (15) but… This reference is nearly 10 years old. Lots of progress have been done since this review.
Line 249-250: I do not agree with this statement. It is not the fact that they both exhibit different mobility that suggest that they do not form heterotetrameric channels. It is the fact that when ORAI1b is expressed either alone or with ORAI1a, it still has the same mobility profile which is different from ORAI1a profile.
299-300: please add reference for the siRNA studies
The paper comes to an end very abruptly. A conclusion paragraph should be added.
English grammatical errors:
Line 16/17: “calcium homeostasis, therefore, physiology and pathophysiology, than originally expected”. This sentence doesn’t make sense
Line 33/34:
“Due to its pleiotropic participation, altered Ca2+ states induce pathological conditions, and even death”. Participation may not be the best word here.
Line 35: a intricated = an intricated
Line 38: Ca2+ retained on intracellular compartment: Calcium isn’t retained on but in the compartments.
Line 51/52: “Those suffering SCID, expressed a homozygous point mutation encoding an R91W replacement”
Line 55: “associated with other diseases, such as autoimmunity”. Autoimmunity is not the right word here. Perhaps autoimmune diseases would be more appropriate.
65: “homolougous”= isoforms or homologues
120: “suffer”
117-123: This sentence is very long and confusing
142: “ It is clear from the beginning”
181: “like in the secretory pathway… and the …”
227: “shown to be associated to its interaction with caveolin”
332: has only recently reported
336: direct = directly
Author Response
Dear Editor and Reviewer,
Thank you for your letter dated 14th November 2022 concerning the above manuscript. We would like to thank the Reviewers for their comments in order to improve the quality of the manuscript. We have modified the manuscript introducing all the requirements and suggestions of the Reviewers. Our responses to the points raised and the modifications made are detailed below.
Yours sincerely
Dr. Juan A. Rosado,
Responses to comments
Reviewer #2:
- Line 58: “the relevance of ORAI2 and ORAI3 remains still unclear (15) but… This reference is nearly 10 years old. Lots of progress have been done since this review.
Answer: We thank the Reviewer for his/her interesting comment. Accordingly, we have modified the section concerning the role of Orai2 and Orai3 isoforms in the modulation of CRAC and their involvement in heteromeric native CRAC channels.
- Line 249-250: I do not agree with this statement. It is not the fact that they both exhibit different mobility that suggest that they do not form heterotetrameric channels. It is the fact that when ORAI1b is expressed either alone or with ORAI1a, it still has the same mobility profile which is different from ORAI1a profile.
Answer: Again, we thank the Reviewer for drawing our attention to this point. In agreement with the Reviewer´s comment, we have rewritten the sentence, that now reads: “The observation that Orai1β exhibits the same mobility profile either expressed alone or co-expressed with Orai1α, which is different from the Orai1α mobility profile, suggest that Orai1α and Orai1β do not form heteromeric channels”.
- 299-300: please add reference for the siRNA studies
Answer: We are not sure which siRNA studies the Reviewer is referring to (around lines 299-300), but we have included the references in the sentence "The Orai1α:Orai1β expression ratio varies among the cell lines investigated but mostly ranging from 0.3 to 1 [85,91], thus suggesting that Orai1 mRNA transduction mostly favors the expression of the short variant.".
- The paper comes to an end very abruptly. A conclusion paragraph should be added.
Answer: Again, we appreciate the Reviewer´s comment. Accordingly, we have added a new section that reads:
- Conclusions
Orai1 is the key pore-forming protein of the CRAC channels, which mediate the prototypical and best-characterized store-operated current ICRAC. In addition, Orai1 promotes store-dependent activation of the ISOC current involving TRPC1. Two variants of Orai1 are expressed at the protein level in mammalian cells. These variants, Orai1 and Orai1, generated by alternative translation initiation, differ in the N-terminal 63 amino acids lacking in the short form, Orai1. While these forms have been reported to support ICRAC with similar efficiencies, their participation in ISOC depends on the cell type investigated and Iarc activation is unique to Orai1. Orai1 variants also differ in their sensi-tivity to fast Ca2+-dependent inactivation, which explains the different Ca2+ signals medi-ated by these forms when expressed individually. The absence of the N-terminal 63 amino acids in Orai1 limits the interaction with different partners, including AC8, AKAP79 or caveolin, as well as serine phosphorylation at residues 27, 31 and 34, present in Orai1. These differences have been proposed to underlie the distinct channel inactivation proper-ties, mobility profiles and NFAT activation mechanisms. The presence of Orai1 and Orai in a given cell type provide an additional tool to generate Ca2+ signals appropriate to the intensity of agonist stimulation.
English grammatical errors:
- Line 16/17: “calcium homeostasis, therefore, physiology and pathophysiology, than originally expected”. This sentence doesn’t make sense
Answer: We thank the Reviewer for his/her corrections. We have rewritten the sentence that now reads: “Furthermore, current evidence supports that abnormal Orai1 expression or function underlies several disorders.”.
- Line 33/34: “Due to its pleiotropic participation, altered Ca2+ states induce pathological conditions, and even death”. Participation may not be the best word here.
Answer: We have rewritten the sentence that now reads: Due to its pleiotropic effects, altered intracellular Ca2+ homeostasis induces pathological conditions [1-3]
- Line 35: a intricated = an intricated
Answer: corrected.
- Line 38: Ca2+ retained on intracellular compartment: Calcium isn’t retained on but in the compartments.
Answer: We thank the Reviewer for his/her correction. We have rewritten the sentene that now reads: where the release of Ca2+ stored in the intracellular compartments…
- Line 51/52: “Those suffering SCID, expressed a homozygous point mutation encoding an R91W replacement”
Answer: We have rewritten the sentence that reads: SCID patients express a homozygous R91W point mutation in the Orai1 protein that impaired T cell activation.
- Line 55: “associated with other diseases, such as autoimmunity”. Autoimmunity is not the right word here. Perhaps autoimmune diseases would be more appropriate.
Answer: We have changed autoimmunity by autoimmune disorders to avoid repetition of diseases.
- 65: “homolougous”= isoforms or homologues
Answer: Thanks, corrected.
- 120: “suffer”
Answer: corrected.
- 117-123: This sentence is very long and confusing
Answer: We have rewritten the sentence that now reads: “Upon cell stimulation by physiological agonists, the reduction in the luminal ER Ca2+ concentration is sensed by STIM1, which suffers a conformational change that allows the activation of Orai1 channels in the plasma membrane [6-8,38].”.
- 142: “ It is clear from the beginning”
Answer: corrected, thanks.
- 181: “like in the secretory pathway… and the …”
Answer: We have rewritten the sentence that now reads: Furthermore, Orai1 might act independently of STIM1 associated to the secretory pathway Ca2+-ATPase-2 (SPCA2) [79,80] or the small conductance Ca2+-activated K+ channel 3 (SK3) [81].
- 227: “shown to be associated to its interaction with caveolin”
Answer: We have modified the sentence that now reads: where Orai1 internalization has been shown to be dependent on its interaction with caveolin.
- 332: has only recently reported
Answer: We have modified the sentence that now reads: “has only recently been reported”
- 336: direct = directly
Answer: Thanks, corrected.